# Effect of Outdoor Cycling, Virtual and Enhanced Reality Indoor Cycling on Heart Rate, Motivation, Enjoyment and Intention to Perform Green Exercise in Healthy Adults

**DOI:** 10.3390/jfmk9040183

**Published:** 2024-10-02

**Authors:** Luca Poli, Gianpiero Greco, Michele Gabriele, Ilaria Pepe, Claudio Centrone, Stefania Cataldi, Francesco Fischetti

**Affiliations:** Department of Translational Biomedicine and Neuroscience (DiBraiN), University of Study of Bari, 70124 Bari, Italy; luca.poli@uniba.it (L.P.); m.gabriele12@studenti.uniba.it (M.G.); ilaria.pepe@uniba.it (I.P.); claudio.centrone@uniba.it (C.C.); francesco.fischetti@uniba.it (F.F.)

**Keywords:** VR, physical activity, leisure time, sedentary, technology, exercise adherence, physical fitness

## Abstract

**Background:** Despite the benefits of physical activity (PA), Italy ranks low in leisure-time PA among European countries. Integrating virtual (VR)/enhanced (ER) reality with exercise equipment could boost PA engagement. Limited studies have explored how VR/ER-integrated cycling activity, compared to outdoor settings, influences PA among university students. Therefore, this study aimed to evaluate the acute effects of a brief cycling session outdoors and indoors on psychological and physiological outcomes, and secondly, investigate the potential of VR/ER-mediated nature experiences as a tool to promote green exercise. **Methods:** In February 2024, thirty-one subjects (20 M and 11 F; age 24.3 ± 3.2 years; BMI 23.5 ± 3.6 kg/m^2^) were involved in this randomized crossover-controlled trial, where they were assigned to three different conditions: ER cycling (ERC), VR cycling (VRC), and outdoor cycling (OUTC). Heart rate (HR), Physical Activity Enjoyment (PACE), Intrinsic Motivation Inventory (IMI), and Intention to Perform Green Exercise (INT-GE) were assessed at the end of each condition. **Results:** The OUTC condition showed significantly greater PACE, IMI, and INT-GE than ERC/VRC (*p* < 0.001), lower HR_mean_ than ERC/VRC (*p* < 0.001 and *p* < 0.05, respectively), and lower HRmax than ERC (*p* < 0.05). **Conclusions:** VRC and ERC enhanced engagement and physiological responses during indoor cycling, but outdoor cycling offered superior benefits in motivation, enjoyment, and future engagement intentions. No significant differences were found between VRC and ERC in promoting intentions for outdoor activities, suggesting both technologies could be equally effective.

## 1. Introduction

Data from the latest European Health Interview Survey (EHIS) position Italy in 21st place out of 27 countries in the ranking of the percentage of adults engaging in leisure-time physical activity (PA): only 26.7% engage in aerobic physical activity at least once a week, whereas the percentage rises to 44.3% among the adult population of Europe as a whole. In Italy, only one in five people engage in aerobic physical activity for at least 150 min per week, thus adhering to the World Health Organization (WHO) guidelines [1], compared to the European average of one in three. Italians are less active than other Europeans even when it comes to active transportation: only 16% use bicycles compared to a European average of 23.6% [2]. University students are at risk of physical inactivity and early development of non-communicable diseases as this population often makes autonomous lifestyle choices for the first time (e.g., PA-related behaviors, dietary choices, etc.) [3,4,5]. Pleasant and sustainable intervention strategies are therefore needed to promote and increase university students’ participation in PA.

The WHO has set ambitious goals for 2030 aimed at promoting physical activity and sustainability. These include a 15% reduction in physical inactivity among adults and adolescents, and the integration of urban policies that encourage active mobility. These initiatives aim to improve public health and environmental sustainability, reflecting a holistic and cross-sectoral approach [6]. Furthermore, the WHO aims to promote people’s ability to exert greater control over their health by embracing active lifestyles while simultaneously preserving the integrity of the environment [7]. Therefore, it is crucial to foster the capability and motivation of individuals, groups, and communities to embrace active and healthy ways of living.

Replacing car travel with cycling can significantly reduce air pollution and reduce exposure to pollutants [8]. Cycling is increasingly recognized as a crucial element in public health guidelines and active transportation policies. However, uncertainties remain about the effectiveness of various intervention strategies to promote cycling [9]. Reduced mortality risk is linked to frequent cycling activity. When compared to not cycling, there was a 17% decreased risk of death for those who cycled for around 100 min a week. The goal should be to increase involvement among those who do not currently cycle frequently in order to maximize the benefits to public health. Moreover, cycling could potentially produce economic benefits for individuals and communities [8]. Given the potential benefits of increased cycling for public health, well-being, the economy, and the environment, investing in effective measures to enhance cycling behavior is essential [8].

Natural environments offer a significant opportunity for promoting health status and well-being, as time spent in nature benefits physical and psychological health [10]. Studies show positive associations between exposure to the natural environment and indicators of physical health, immune function, obesity, and the incidence of other chronic diseases [11,12,13]. It has been suggested that to maintain higher levels of overall health and well-being, people should spend at least 120 min per week in nature [14]. Green exercise (i.e., PA undertaken in the presence of a natural environment) is particularly beneficial in this regard, as it combines the benefits of physical activity with those provided by exposure to nature [15]. The integration of new technologies, such as virtual reality (VR) and enhanced reality (ER), with traditional exercise equipment (e.g., stationary bikes) could help promote participation in PA, especially among young adults [16,17]. Moreover, a recent study found that brief physical activity breaks performed in outdoor (OUT) settings or throughout exergame modalities can improve selective attention and executive functions [18].

Virtual green exercise, which combines PA with virtual nature representations, is designed to offer the benefits of nature exposure and exercise to those who cannot access to natural environment [19]. This concept has gained interest in the medical community for its potential health benefits [14]. Research by Calogiuri et al. [20] suggests that virtual green exercise can enhance exercise output and improve mood, relieve stress, and aid cognitive restoration. However, the research is still nascent, and a review by Lahart et al. [21] found no consistent psychological or physiological benefits compared to indoor exercise, highlighting the need for more rigorous studies.

VR offers a highly immersive experience, making it a promising tool for simulating natural environments and providing health benefits [22,23]. Immersion may be crucial for these benefits, and VR technology is increasingly used in medical care [24,25]. There are two main types of VR simulations: computer-generated scenarios and 360° videos of real scenes [26]. Interactive designs, such as games and physical activities, can enhance the virtual nature experience, although the effects may vary with different VR simulation types.

Cybersickness is a known issue with VR, but its impact and prevention strategies are under-researched [27,28]. The extent to which adverse symptoms might negate the benefits of virtual nature and how different immersion levels or environments contribute to cybersickness remain unclear, making it difficult to generalize the benefits of virtual nature [28].

ER, also called non-immersive virtual reality, is a two-dimensional scenario delivered through a screen and controlled, in several specific aspects, through a device or components of an instrument that incorporate it [29].

VR and ER technologies have been identified as a novel strategy for encouraging PA and promoting healthier behaviors [30]. Physical exercise can often be influenced by external factors such as weather conditions, lighting, and traffic [31]. However, by incorporating VR and ER into exercise routines, it is possible to mitigate these negative environmental impacts and improve exercise motivation. Consequently, VR and ER technologies have the potential to complement physical interventions and modify human behaviors.

In modern societies, a small percentage of adults report exercising at a level that complies with the majority of public health recommendations [32], indicating that a large number of people lack the motivation necessary to engage in appropriate levels of physical activity. There are two general classes of factors that can account for lack of motivation: low perceived competence or low interest. Some of the qualitative aspects of motivation, such as the degree of perceived autonomy or internal locus of causality, are at least somewhat related to the stability of that motivation [33]. In other words, a significant portion of the population either lacks the motivation or does not have the incentive to engage in PA, or they are motivated by external factors that cannot result in prolonged engagement. This emphasizes how important it is to pay closer attention to the objectives and self-control traits linked to consistent exercise and PA [34].

According to feedback theory, emotions serve the function of providing feedback on which behaviors should be pursued or avoided in the future [35]. In this context, anticipated emotions are considered more important in guiding behavior than emotions actually experienced. This, in the context of exposure to nature through VR/ER, can be facilitated by high levels of presence, i.e., the (psychological) sense of being in the virtual environment. Some studies [19,20] have proposed VR/ER as an effective tool in behavioral change interventions, particularly in promoting outdoor PA.

Currently, few studies have compared the effects of exercise integrated with VR, particularly ER, to those performed in outdoor settings on the promotion of PA in a population of healthy university students. Even fewer have investigated such effects using cycling activities. The primary aim of this study was to investigate the effects of three types of brief physical activity (cycling) on parameters related to intrinsic motivation, enjoyment, and heart rate (HR) variations in young adult Italian university students. Secondly, we aimed to explore the potential of these activities as a tool to promote outdoor physical exercise. We hypothesize that the three conditions (VR, ER, and OUT) will result in similar levels of intrinsic motivation, enjoyment, and HR variation. Regarding the secondary outcome, we hypothesize that, among the technological conditions, the VR condition will induce greater transfer compared to the ER condition in the intention to engage in outdoor physical activities.

## 2. Materials and Methods

### 2.1. Participants

Participants were recruited from the local fitness center, Polignano a Mare, Bari, Italy (Living S.S.D.), selecting only university students aged between 18 and 30 years. We considered the following exclusion criteria: (1) declining to take part in the study, (2) indications of musculoskeletal disorders or significant lower limb injuries, (3) acute or chronic disease, (4) taking medications that may affect mood and cognitive regulation, (5) habitual performance of physical/sporting activity in an outdoor setting, (6) prior experience in using ER-cycling and/or VR-cycling technologies.

So, thirty-one healthy adults (age, 24.3 ± 3.2 years; body mass, 70.8 ± 14.9 kg; body height, 172.9 ± 9.5 cm; BMI, 23.5 ± 3.6 kg/m^2^; sex, 11 females and 20 males) were enrolled in this experimental work (between February and March 2024).

A preliminary power analysis [36], was conducted with a Type I error rate α = 0.05 and a Type II error rate β = 0.10 (indicating a statistical power of 90%, i.e., 1 − β). The analysis determined that a total of 27 participants would be adequate to detect medium effect sizes “within factors” (f = 0.30). Informed consent was collected, including comprehensive details about the study’s procedures and tests, along with information about their right to withdraw at any time. This research was carried out in compliance with the Declaration of Helsinki and received approval from the Ethics Committee of Bari University (protocol code 0015637|16 February 2023).

### 2.2. Study Design

A randomized controlled crossover study utilizing within-subjects repeated-measures design was conducted. Each participant experienced all conditions in a random and counterbalanced sequence.

Participants were randomly allocated to one of three cycling conditions: enhanced reality cycling (ERC), virtual reality cycling (VRC), or outdoor cycling (OUTC). The randomization was executed using Research Randomizer, a tool available on an official public website (www.randomizer.org, accessed in February 2024). To reduce potential circadian-related effects, all conditions and assessments took place between 9:30 a.m. and 12:30 p.m., with a one-week washout period between trials. Participants were instructed to abstain from intense physical activity and caffeine for 24 h before each session and throughout the data collection period.

Researcher 1, who was blinded to the treatment assignments, assessed subject characteristics and all outcome measures following each cycling condition. The interventions and evaluation were carried out in the same outdoor setting and indoor fitness center located in Polignano a Mare, Bari (Italy). Researcher 2, responsible for conducting the interventions, did not participate in the assessments and was unaware of the study’s objectives or the treatment allocations.

Figure 1 presents a flow chart illustrating the randomized allocation of participants across the three conditions.

### 2.3. Experimental Protocol

Participants were instructed to refrain from engaging in any vigorous activities for a minimum of 30 min before the intervention. Following this, each participant carried out their designated cycling condition based on random assignment.

For the VRC condition, the identical cycling route (Figure 2) used in the OUTC condition was filmed with the Insta360 X3 (Arashi Vision Inc., Shenzhen, China) to produce a 5 K 360-degree video. During the VRC sessions, participants used VR headsets with integrated headphones, along with a Samsung Galaxy S20 FE phone (Samsung Electronics Co., Ltd, Suwon, South Korea) inserted into the headsets, offering an immersive VR simulation of the recorded outdoor cycling route. The 360-degree footage was captured at the same speed (14.5 km/h) as used during indoor sessions. This speed falls into the average speed for recreational cycling activities [37].

In all conditions, the exposure duration was set to 10 min. This time span was selected because it has been previously identified as the minimum necessary to achieve health and well-being benefits from both nature exposure [38] and PA [39]. Additionally, this duration helps prevent or limit the onset of cybersickness [40,41].

The OUTC condition was performed by cycling outside on an outdoor bike pathway (Figure 2). Each participant cycled for 10 min at a predetermined speed of 14.5 km/h. The OUTC condition was performed on a regular muscular city bike, and participants were instructed to maintain the established speed, which was displayed in real time through a cyclo-computer (Garmin^®^, Edge 540; Garmin Ltd., Olathe, KS, USA) [42]. The ERC condition was performed on a stationary bike (Technogym^®^, smart bike; Technogym S.p.A, Cesena, Italia) at 14.5 km/h speed, equipped with a monitor (22″ full-HD touchscreen) that reproduces a natural landscape in first person. During the cycling experience, the surrounding landscape adapts to the speed in real time. The same resistance level was set for all three conditions (VRC, OUTC, and ERC) to ensure consistency in the physical effort required across different cycling modalities.

Immediately after the end of each cycling condition, lasting 10 min each, participants were measured for HR (average and maximum) and subjective ratings of intrinsic motivation, physical activity enjoyment, and intention to perform green exercise by answering the Intrinsic Motivation Inventory (IMI), the Physical Activity Enjoyment Scale (PACES), and Intention to Perform Green Exercise (INT-GE) questionnaires, respectively.

### 2.4. Measures

#### 2.4.1. Heart Rate

Throughout the cycling sessions, the participant’s heart rates were consistently tracked with a wearable device (Polar^®^, Verity Sense; Polar Electro Oy, Kempele, Finland). Post-test, both the average heart rate (HR_mean_) and the maximum heart rate (HR_max_) were documented. Prior research [43,44] has confirmed the accuracy of such devices for heart rate monitoring in adults. HR_mean_ and HR_max_ are measurements that are relatively simple to obtain and analyze, and also a useful indicator of cardiovascular load during physical exercise. HR_mean_ provides an idea of the sustained cardiac load throughout the entire exercise session, while HR_max_ indicates the peak effort reached. These data could help in understanding the physical intensity and the level of cardiac stress imposed by the different cycling conditions (ERC, VRC, OUTC).

#### 2.4.2. Intrinsic Motivation Inventory (IMI)

The Intrinsic Motivation Inventory (IMI) is a multidimensional measurement tool intended to assess participants’ subjective experience related to a target activity. It has been used in several experiments related to intrinsic motivation and self-regulation, originating from original studies [45,46]. The IMI has been widely used in research to assess intrinsic motivation in different scientific fields, such as the academic, and has been gaining acceptance in the sport and exercise domain [47,48]. A smaller number of IMI items can be selected and modified depending on the activity and research question, without adversely affecting the psychometric properties of the measure. Overall, the IMI is a very flexible instrument that offers the opportunity to select/modify relevant items to assess intrinsic motivation in any sport/exercise setting [49]. The used scale consists of 25 items, including the three subscales of value/usefulness (Cronbach α 0.87), interest/enjoyment (Cronbach α 0.94), and perceived choice (Cronbach α 0.89). Following each condition, participants evaluate their level of agreement with each statement using a seven-point scale, from 1 to 7, where 1 means not at all true, 4 is somewhat true, and 7 is very true.

It is designed to be filled out right after engaging in a PA, offering insights into individuals’ intrinsic motivation to perform that specific activity.

#### 2.4.3. Physical Activity of Enjoyment Scale (PACES)

The Physical Activity Enjoyment Scale (PACES) is a survey designed to gauge how much a person enjoys engaging in physical activity [50,51]. The questionnaire includes 16 statements rated on a 5-point Likert scale, ranging from 1 (strongly disagree) to 5 (strongly agree). Among these, nine statements reflect positive feelings (e.g., “it energizes me”), while seven express negative sentiments (e.g., “it’s boring”), with Cronbach’s alpha values between 0.78 and 0.89 [52]. The PACES evaluates several aspects of enjoyment, such as positive emotions, psychological involvement, and contentment with the activity [53,54]. It is frequently utilized in studies to understand people’s views and attitudes towards physical exercise, offering insights into the motivational elements that affect exercise behaviors.

#### 2.4.4. Intention to Perform Green Exercise (INT-GE)

Intention to Perform Green Exercise (INT-GE) is part of a group of three questionnaires developed to assess individuals’ beliefs about green (outdoor) exercise [55].

This questionnaire is used to assess the effects of exposure to different conditions on participants’ overall Intention to Perform Green Exercise [56].

The tool comprises five items (i.e., “I want to do green exercise”), each evaluated on a scale from 1 to 7, with 1 meaning “Absolutely disagree” and 7 meaning “Absolutely agree.” Before the items were presented, a definition of green exercise (any type of physical activity or exercise performed outdoors in natural environments like urban parks, beaches, cliffs, countryside, forests, etc.) was provided. Each item specifies a timeframe (“over the next week”) (Cronbach alpha 0.92).

### 2.5. Statistical Analysis

Statistical analysis was performed utilizing JASP software version 0.17.2.1 [57]. Data were reported as mean (M) values with standard deviations (SD) and tested for sphericity using Mauchly’s test. When sphericity was not met, the Greenhouse–Geisser correction was applied.

The Shapiro–Wilk test assessed the normality of all variables. Differences between the three conditions were examined using a one-way repeated measures ANOVA. If significant differences were found, post hoc tests with Bonferroni correction were used to determine specific significant comparisons.

Eta squared (η2) was used to measure the effect size within groups, classified as small (η2 < 0.06), moderate (0.06 ≤ η2 < 0.14), and large (η2 ≥ 0.14) [58]. For post hoc tests, Cohen’s d was calculated, with the effect sizes interpreted as small (0.20 ≤ d < 0.50), moderate (0.50 ≤ d < 0.79), and large (d ≥ 0.80) [59]. Statistical significance was set at *p* ≤ 0.05.

## 3. Results

All thirty-one participants involved in the study underwent each of the three cycling conditions without injuries during the research period. The values for the three cycling conditions are detailed in Table 1.

One-way ANOVA with repeated measures found significant “within-subjects effects” for all the outcomes measures: HR_mean_ (F = 11.701, *p* < 0.001, η2 = 0.281, large ES), HR_max_ (F = 3.617, *p* = 0.033, η2 = 0.108, moderate ES), IMI-Interest/enjoyment (F = 20.179, *p* < 0.001, η2 = 0.402, large ES), IMI-Value/usefulness (F = 16.669, *p* < 0.001, η2 = 0.357, large ES), IMI-Perceived choice (F = 7.373, *p* < 0.001, η2 = 0.197, moderate ES), PACES (F = 12.083, *p* < 0.001, η2 = 0.287, large ES), and INT-GE (F = 7.613, *p* = 0.001, η2 = 0.202, large ES).

Bonferroni post hoc test showed that HR_mean_ was significantly higher during the ERC (t = 4.743, *p* < 0.001, d = 0.627, moderate ES) and VRC (t = 3.196, *p* = 0.007, d= 0.422, small ES) compared to OUTC sessions.

Greater HR_max_ was found only in the ERC compared to OUTC sessions (t = 2.575, *p* = 0.038, d = 0.339, small ES).

Intrinsic motivation level, measured by IMI, was significantly higher in the OUTC condition compared to the others, in the interest/enjoyment (OUTC vs. ERC: t = −5.966, *p* < 0.001, d = 1.061, large ES; OUTC vs. VRC: t = −4.872, *p* < 0.001, d = 0.867, large ES) and value/usefulness (OUTC vs. ERC: t = −4.415, *p* < 0.001, d = 0.715, moderate ES; OUTC vs. VRC: t = −5.430, *p* < 0.001, d = 0.879, large ES) subscales, while for the perceived choice subscale the values were significantly higher for the OUTC than for the ERC session (OUTC vs. ERC: t = −3.774, *p* = 0.001, d = 0.464, small ES) and for VRC compared to ERC (t = −2.501, *p* = 0.045, d = 0.308, small ES).

The level of enjoyment measured by PACES was significantly higher in the OUTC session compared to the others (OUTC vs. ERC: t = −4.081, *p* < 0.001, d = 0.778, moderate ES; OUTC vs. VRC: t = −4.414, *p* < 0.001, d = 0.841, large ES).

Finally, the INT-GE score was significantly higher in OUTC compared to the other conditions (OUTC vs. ERC: t = −3.622, *p* = 0.002, d = 0.580, moderate ES; OUTC vs. VRC: t = −3.068, *p* = 0.010, d = 0.492, small ES).

## 4. Discussion

The present study aimed to investigate the effects of three different cycling conditions: ERC, VRC, and OUTC, on intrinsic motivation, enjoyment, and stress among university students. Additionally, it explored the potential of VR/ER-mediated naturalistic experiences in promoting outdoor physical exercise. Our findings showed significant differences across the three conditions, shedding light on the relative effectiveness of each in fostering PA engagement and enjoyment in the university student population.

Firstly, we observed a higher HR_mean_ in the ERC and VRC conditions compared to the OUTC, contrary to what has been observed in previous similar work [60]. This difference could be explained by the different types of PA performed, walking versus cycling, and the different times of the year in which the experimental intervention was set up, summer versus winter; in the previous study, a high outdoor temperature [21] may have affected the increase in the HR_mean_. Previous studies, albeit with different physical activities and modalities, show either no difference between activities performed outdoors versus indoors [61], or higher heart rate values for activities performed outdoors versus indoors, despite similar perceived exertion [62,63].

However, our results suggest that indoor cycling with enhanced or virtual environments may induce greater physiological responses, and probably greater perceived exertions. The higher HR_max_ found in the ERC compared to OUTC sessions further supports this notion. This is in line with other work that has observed a reduction in heart rate (mean and max) during PA performed in outdoor settings [21,64]. Moreover, the difference between VRC and OUTC was not significant for the HR_max_, suggesting that VR might provide a less strenuous exercise environment than ER. However, it is important to note that several previous studies observe a reduction in heart rate values after exposure to a VR experience in general and not specifically related to exposure to a virtual natural environment [65,66,67,68,69].

Intrinsic motivation, which may be an indicator of future PA behaviors and adherence [70], was significantly higher in the OUTC condition across all subscales (interest/enjoyment, value/usefulness, and perceived choice). This aligns with the broader literature highlighting the benefits of natural environments on psychological well-being and motivation (Twohig-Bennett and Jones, 2018; White et al., 2019) [10,11]. The outdoor setting appears to offer unique motivational benefits that are not entirely replicated by VR or ER, even when these technologies simulate natural environments. The higher perceived choice in the VRC compared to the ERC also suggests that participants felt they had more autonomy and control over their actions, which is an important aspect of intrinsic motivation. When people feel that they are acting out of their own volition and not because of external pressures, their intrinsic motivation is generally higher [47,71].

Enjoyment is a key predictor of exercise adherence [72], and our results show that the PACES scores were significantly higher in the OUTC session, in accordance with previous studies [73,74,75], indicating greater enjoyment of outdoor cycling compared to indoor cycling with VR or ER. This finding underscores the importance of real-world natural environments in enhancing the enjoyment of recreational physical activities, which is crucial for sustaining long-term active behaviors [15].

Over the past three decades, research has consistently shown that interactions with nature offer a wide range of health and well-being benefits [14,76,77]. Various pathways have been identified linking natural contact to physical and psychological health, such as air quality improvement, heat modulation, PA, social cohesion, and stress reduction [78,79].

Green exercise, which includes activities like visiting naturalistic locations, outdoor recreation, and exercising or walking in natural environments, provides synergistic benefits from nature contact and PA. Regular green exercise can lead to long-term benefits such as better self-rated health, higher life satisfaction, and stronger social support. Multiple review studies confirm the additional health and well-being advantages of green exercise compared to indoor or urban PA [80,81].

Our results show that direct exposure to outdoor environments might more effectively promote future engagement in outdoor physical activities; in fact, INT-GE scores were significantly higher following the OUTC sessions compared to both ERC and VRC. While VR and ER can offer immersive and stimulating experiences, they may not sufficiently replicate the motivational impact of actual outdoor exercise, a critical consideration for interventions aimed at promoting sustained PA. Moreover, contrary to our initial hypothesis, we found no significant differences between VRC and ERC conditions in promoting intention to engage in outdoor physical activities, suggesting both technologies could be equally effective.

Our findings have several practical implications. While VR and ER technologies offer innovative ways to engage individuals in PA, especially in settings where access to natural environments is limited, they should complement rather than replace outdoor activities. For college students, experiencing recreational PA could contribute to maintaining an active lifestyle in the long run. Incorporating more outdoor exercise opportunities into their routine could improve their physical and psychological well-being. Universities should consider developing programs that encourage outdoor physical activities, taking advantage of the intrinsic benefits of motivation and enjoyment associated with natural environments. These efforts could contribute to a holistic approach to sustainability, emphasizing both human health and the well-being of the planet.

### Strength and Limitations

As far as we know, this is the first study that compares the effects of PA integrated with VR and ER with that performed outdoors on promoting PA in a population of healthy college students. In addition, we examined these effects using cycling activities. Moreover, the outdoor activity was conducted during the winter season, in contrast to many similar studies that typically take place during the spring or summer. This enhances the validity of our findings, as winter conditions often pose challenges to outdoor PA, whereas spring and summer conditions generally promote it.

Highlighting the effects of different cycling modalities, it encourages regular physical activity, which not only enhances individual health but could also reduce healthcare costs over time. Utilizing natural environments for exercise supports the preservation and use of green spaces, benefiting environmental conservation efforts. Additionally, the findings could suggest practical implications for urban planning and university programs and find new ways to motivate young people to participate in physical activity. Moreover, active transportation can contribute to reducing carbon emissions and improving air quality, thus addressing broader environmental sustainability goals. Ultimately, the study shows initial insight into enjoyable and motivating outdoor activities like cycling that can lead to sustained physical activity, supporting both personal health and the well-being of society.

However, there are some limitations to this study. The sample size was relatively small and consisted of students from a specific geographic area, which may limit the generalizability of the findings.

We only measured and assessed HR (mean and maximal), without directly assessing perceived exertion (RPE). This may have limited the interpretations of the data, since HR does not always reflect perceived exertion (e.g., HR might be higher but perceived exertion might be low) [82]. Furthermore, HR was not represented as a percentage of everyone’s maximum HR, which reduces its accuracy as an indicator of exercise intensity; this is also because the determination of maximum HR requires a very intense and strenuous effort that could have limited participation in the study. Hypothetical maximal HR related to age was not utilized due to the constraints of these formulas [83].

While the scenario displayed during the VRC session was the same one to which participants were exposed in the OUTC session, the scenario shown during the ERC session, though similar, was different. This could somehow represent a bias of the study, notwithstanding that the results obtained in the two virtual conditions (ERC and VRC) were similar.

The acute intervention study design offers preliminary insights into the specific forms of recreational PA that may enhance intrinsic motivation, potentially leading to sustained engagement in these activities over time. However, to validate and extend these initial findings, longitudinal studies with subsequent follow-up are required.

Additionally, the controlled study environment may not accurately represent participants’ real-world experiences.

## 5. Conclusions

In conclusion, while VR and ER technologies show promise in enhancing engagement and physiological responses during indoor cycling, outdoor cycling offers superior benefits in terms of intrinsic motivation, enjoyment, and intention to engage in future outdoor activities. Promoting outdoor physical activities remains crucial for enhancing the overall health and well-being of university students. Our results show the potential effect of incorporating outdoor physical activity to promote sustained engagement and well-being. However, we cannot provide specific recommendations due to the acute nature of our intervention.

Future research should explore the long-term effects of these conditions and include diverse populations. Additionally, investigating other forms of physical activity beyond cycling could provide a broader understanding of how different environments impact various types of exercise.

## Figures and Tables

**Figure 1 jfmk-09-00183-f001:**
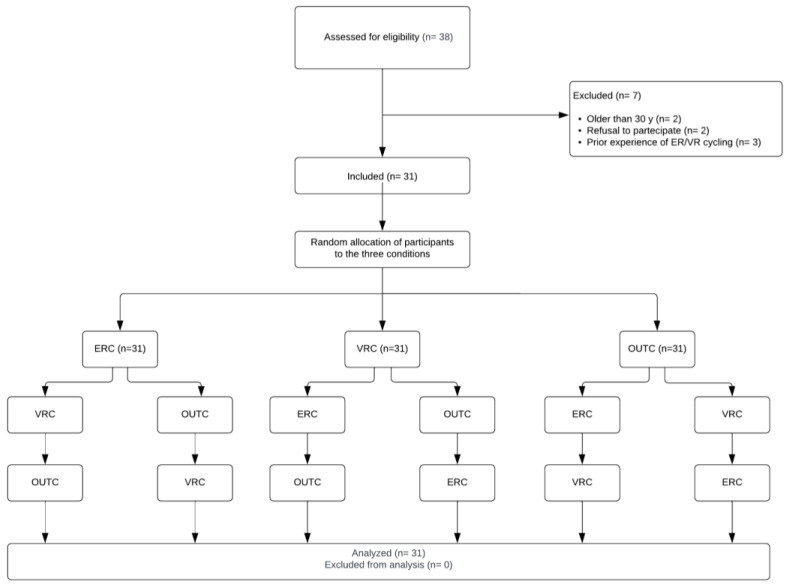
Randomized allocation to the different conditions.

**Figure 2 jfmk-09-00183-f002:**
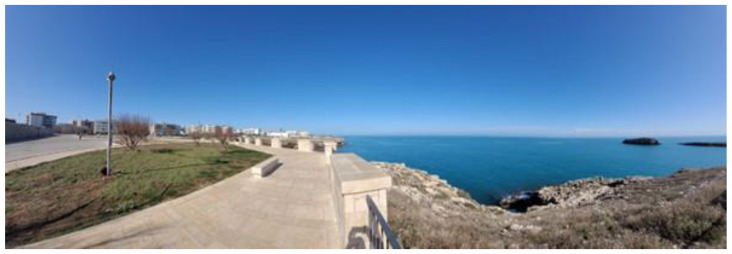
Outdoor cycling route for OUTC/VRC conditions.

**Table 1 jfmk-09-00183-t001:** Changes found between the three cycling conditions.

Variables	ERC	VRC	OUTC
HR_mean_ (bpm) HR_max_ (bpm)	111.3 ± 14.1 ^a^*** 124.3 ± 16.7 ^a^*	108.4 ± 13.0 ^b^** 119.9 ± 16.1	102.4 ± 15.3 ^a^***^,b^** 118.5 ± 18.4 ^a^*
IMI-Interest/enjoyment (scores)	4.9 ± 0.9 ^a^***	5.0 ± 0.9 ^b^***	5.8 ± 0.7 ^a^***^,b^***
IMI-Value/usefulness (scores)	5.3 ± 0.9 ^a^***	5.1 ± 1.3 ^b^***	6.0 ± 0.9 ^a^***^,b^***
IMI-Perceived choice (scores)	5.9 ± 0.8 ^a^**^,c^*	6.2 ± 0.8 ^c^*	6.3 ± 0.8 ^a^**
PACES (scores)	64.8 ± 6.6 ^a^***	64.2 ± 10.5 ^b^***	71.0 ± 6.6 ^a^***^,b^***
INT-GE (scores)	4.7 ± 1.0 ^a^**	4.8 ± 1.2 ^b^**	5.4 ± 1.3 ^a^**^,b^**

Data are reported as mean ± SD. Abbreviations: ERC, enhanced reality cycling; VRC, virtual reality cycling; OUTC, outdoor cycling; HR, heart rate; IMI, Intrinsic Motivation Inventory; PACES, Physical Activity Enjoyment Scale; INT-GE, Intention to Perform Green Exercise. ^a^ Significant difference between ERC and OUTC (*p* < 0.05); ^b^ significant difference between VRC and OUTC (*p* < 0.05); ^c^ significant difference between VRC and ERC (*p* < 0.05). * *p* < 0.05; ** *p* < 0.01; *** *p* < 0.001.

## Data Availability

The data presented in this study are available on request from the first author. The data are not publicly available due to privacy.

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
