# Peer review of "Effect of Outdoor Cycling, Virtual and Enhanced Reality Indoor Cycling on Heart Rate, Motivation, Enjoyment and Intention to Perform Green Exercise in Healthy Adults"

_jfmk, 2024, doi:10.3390/jfmk9040183_

Round 1
Reviewer 1 Report
Comments and Suggestions for Authors
The Authors explored the effect of outdoor vs indoor cycling activities in a sample of young subjects. The indoor cycling was also explored in a virtual reality (VR) and enhanced reality (ER) conditions. A randomized crossover controlled trial was conducted to analyze the within-effect of conditions. During the cycling sessions the heart rate (HR) was monitored. At the end of each session the mean and maximum of HR, the intrinsic motivation, the intention to perform and the enjoyment to perform the cycling activities proposed were evaluated with surveys to participants. Results showed that VR and ER conditions had similar benefits, however the outdoor cycling had better physiological responses and intention and motivation to perform cycling activities post-experiment.
The Authors well introduced the need of their work in the background section where the promotion of green exercise is in line with the WHO guidelines and ecological issues.
The limitations of the study have been acknowledged in the manuscript (small sample size).
Figures and table provided appear suitable but could be improved as detailed below.
The manuscript is generally well written and can be of interest for the readers of this Journal. I have only minor comments that I resume below.
Minor comments:
Abstract
Line 17: 31 participants have been recruited, but in the following parentheses the sum of males and females is 32. Also, in the figure of allocation, 32 participants have been declared. Please, consider changing the correct number of participants throughout the manuscript.
Introduction
Line 78 and throughout the background section: the outdoor and virtual cycling have been explained; however, the ER condition is not clear. Please, provide more information about this condition in the introduction section.
Materials and Methods
Line 192: consider removing “g” from “Garming”.
Lines 263-264: consider adding a reference to justify the classification of eta squared as the Authors already did for Cohen’s d in the next sentence (ref. n. 51).
Results
In order to make the results section clearer and more readable, consider removing from the paragraph the statistic values and add them in the table. Also, in the table, it could be added a column for p-values, written in extenso and not with asterisks, and effect size (Cohen’s d and eta squared). Only main results could be left in the result section.
Line 277: HR mean should be written as subscript
References
Some references are written in bracket squares and numbers, as required, some others are as “Surname et al., year of publication”. Please, revise the in-text citation and reference list according to the Journal guidelines.
Reviewer 2 Report
Comments and Suggestions for Authors
Ms. No.: jfmk-3176849
Title: Effect of Outdoor Cycling, Virtual and Enhanced Reality Indoor Cycling on Heart Rate, Motivation, Enjoyment and Intention to Perform Green Exercise in Healthy Adults
Reviewer 1 comments
This manuscript describes a well-designed and executed study aimed at the evaluation of the motivational factors related to virtual and enhanced reality (VR and ER, respectively), and their translation towards activities outdoors. Cycling is the researcher's “weapon of choice” since it is a fitness technology that has matured recently and has become a standard in fitness centers all across the globe. Similarly, many gadgets can help obtain accurate physio and other data, and this research team took full advantage of this existing technology. Findings are sound and supported by collected data, and there is enough room to grow this line of research by expanding the sample size and inclusion of other parameters. There are only minor issues with the presentation of the body of knowledge, which the authors should quickly address.
One aspect not addressed in this study is the daily burden related to professional (work, study) and personal (family) activities and how they impact the inhibition of PA. Modern technology (smart watches) could help obtain precise, quantifiable measures and find the threshold values for motivation.
Specific comments
Line 78: “Virtual green exercise, which combines PA with virtual nature representations, is designed to offer the benefits of nature exposure and exercise to those who cannot access real nature [16].” Would you mind providing some details about the criteria used to make this determination? At face value, this is a “strange” statement since no VR can be substituted for “real nature.” The term “real nature” is also somewhat odd, considering nature is nature, and only substitutes need identification via additional descriptors, the “real” in this case.
Line 86: “VR offers a highly immersive experience, making it a promising tool for simulating natural environments and providing health benefits [19,20].” The level of VR-immersion likely also depends on the type of display technology used, i.e., goggles vs LCD monitor. You started this paragraph with a promise to explore the advantages of VR and the effect of photorealism (computer-generated vs real-world videos) on the “immersivity,” but it falls short on the detail. Are there any studies where these effects were quantified, or is it application-driven?
Line 98: “In contemporary countries, a small percentage…” “Modern societies” is probably a better descriptor.
Line 109: “According to [29] feedback theory, emotions serve the function of providing feedback…” The reference marker can be placed at the end of the sentence without changing the meaning of it.
Line 117: “… the promotion of PA (intrinsic”. If you remove this (the promotion of PA) fragment and drop the parentheses, it would make a sentence with a good flow of thought.
Line 124: “that the three conditions (VR, ER, and OUT).” The abbreviation OUT is not explained (in the previous sentence) unless it should be OUTC (?).
Line 131: “Thirty-one healthy adults was enrolled…” It’s plural, so WERE enrolled.
Line 132: “Participants were recruited from the local fitness center, Polignano a Mare, Bari (Italy), selecting only university students aged between 18 and 30 years.” This sentence makes for a much better intro to your study design. Move up.
Line 140: “A preliminary power analysis [30], was conducted with a Type I error rate of 0.05 140 and a Type II error rate of 0.10 (indicating a statistical power of 90%).” The G*Power software calls it a priori power analysis, and it depends on the assumed effect size (d). In this case, I used d=0.5 to get numbers similar to yours with α = 0.1 and Power (1 – β) of 0.90. These parameters should be only reported since if you decrease the α to 0.05, the sample size will increase to 34. There is no need to mention the types of errors if the two-tailed error distribution is assumed, which is a safe bet in any scenario.
Line 175: “For the VRC condition, the identical cycling route (Figure 2) used in the OUTC…” Could you also provide a map of the course, potentially with the elevation diagram? These are standard in apps used for cycling tracking like Strava.
Line 318: “However, our results suggest that indoor cycling with enhanced or virtual environments may induce greater physiological responses,…” One significant difference you need to remember in this context is the key difference between indoor and outdoor cycling. That difference is one needs to keep pedaling all the time (continuously) indoors, which is not the case outdoors. The cadence sensor would help you have insight into that, and these are affordable.
Comments on the Quality of English LanguageIncluded in the "Comments and Suggestions for Authors
" above.
Reviewer 3 Report
Comments and Suggestions for Authors
The paper proposed Integrating virtual (VR)/enhanced (ER) reality with exercise equipment. However, the paper seems to miss the fundamental basis of the experiment. Only one figure, Figure 2, reflects the experimental condition. However, I cannot find the people cycling in Figure 2. The paper is quite short and doesn't explain the motivation, research method, and result properly.
